# Dual-Functional Polymeric Micelles Co-Loaded with Antineoplastic Drugs and Tyrosine Kinase Inhibitor for Combination Therapy in Colorectal Cancer

**DOI:** 10.3390/pharmaceutics14040768

**Published:** 2022-03-31

**Authors:** Ying-Hsia Shih, Cheng-Liang Peng, Ping-Fang Chiang, Ming-Jium Shieh

**Affiliations:** 1Isotope Application Division, Institute of Nuclear Energy Research, Taoyuan 32546, Taiwan; shihys@iner.gov.tw (Y.-H.S.); ckdopamine@iner.gov.tw (P.-F.C.); 2Institute of Biomedical Engineering, College of Medicine and College of Engineering, National Taiwan University, Taipei 100, Taiwan; 3Department of Oncology, National Taiwan University Hospital and College of Medicine, Taipei 100, Taiwan

**Keywords:** SN-38, sunitinib, polymeric micelles, colorectal cancer

## Abstract

The aim of this research was to evaluate the receptor tyrosine kinase inhibitor Sunitinib combined with SN-38 in polymeric micelles for antitumor efficacy in colorectal cancer. First, SN-38 and Sunitinib co-loaded micelles were developed and characterized. We studied cell viability and cellular uptake in HCT-116 cells. Then, subcutaneous HCT-116 xenograft tumors were used for ex vivo biodistribution, antitumor efficacy, and histochemical analysis studies. Results of cellular uptake and ex vivo biodistribution of SN-38/Sunitinib micelles showed the highest accumulation in tumors compared with other normal organs. In the antitumor effect studies, mice bearing HCT-116 tumors were smallest at day 28 after injection of SN-38/Sunitinib micelles, compared with other experiment groups (*p* < 0.01). As demonstrated by the results of inhibition on multi-receptors by Sunitinib, we confirmed that SN-38/Sunitinib co-loaded micelles to be a treatment modality that could inhibit VEGF and PDGF receptors and enhance the antitumor effect of SN-38 (*p* < 0.05). In summary, we consider that this micelle is a potential formulation for the combination of SN-38 and Sunitinib in the treatment of colorectal cancer.

## 1. Introduction

Colorectal cancer (CRC) is the third most common cancer in the world and has a poor prognosis and a high mortality rate [1,2]. Chemotherapy is effective but is often accompanied by poor efficacy and severe side effects [3,4]. There are many various microscale or nanoscale materials such as microemulsions [5,6,7,8], liposomes, micelles, and nanoparticles used as drug delivery systems in cancer treatments. Methoxy poly-(ethylene glycol)-poly(ε-caprolactone) (mPEG-PCL) is useful and one of the most promising nanomaterials for cancer chemotherapeutics [9,10]. Many receptor tyrosine kinase inhibitors (RTKIs) have been shown to alter the tumor microenvironment [11,12,13]. Sunitinib is an oral multi-targeted receptor tyrosine kinase inhibitor (RTKI) that is commonly used for the treatment of gastrointestinal stromal tumor (GIST), advanced renal cell cancer, and pancreatic cancer. Therefore, we applied Sunitinib as a therapeutic in colorectal cancer, because Sunitinib targets include receptors for platelet-derived growth factor receptors (PDGF-α and PDGF-β receptors) and vascular endothelial growth factor receptors (VEGF-1, VEGF-2 and VEGF-3 receptors) that play a role in both tumor angiogenesis and tumor cell proliferation. Additionally, Sunitinib also inhibits the stem-cell factor receptor (SCF receptor), Fms-like tyrosine kinase-3 receptor (FLT-3 receptor), and colony stimulating factor-1 receptor (CSF-1 receptor) [13,14,15,16]. Moreover, Sunitinib (Sunitinib malate, Sutent^®^ capsules; Pfizer, New York, NY, USA) is a Food and Drug Administration (FDA) approved antitumor drug [17]. Thus, it is possible that Sunitinib will enhance the antitumor effect by inhibiting multiple targeted receptors.

Camptothecin-11 (CPT-11 also known as Irinotecan) is a water-soluble derivative for clinical use [18]. It is a cytotoxic alkaloid that was first isolated from Camptotheca acuminate [19,20], and causes cell death via inhibition of topoisomerase I (Top I) [21,22]. However, SN-38 (7-ethyl-10-hydroxy-camptothecin), the active metabolite of CPT-11, has 100-fold to 1000-fold higher cytotoxicity than CPT-11 [23,24]. Therefore, SN-38 has excellent anticancer effects. As SN-38 is often used at high doses that lead to side effects in patients, the combined delivery of SN-38 and Sunitinib through copolymer micelles can be expected to increase drug circulation time, reduce the required effective dose and associated side effects, and enhance the antitumor effect.

Therefore, this study investigated whether Sunitinib combined with SN-38 could enhance antitumor effect in colorectal cancer when delivered through nanoscale micelles (Figure 1). With this objective, we prepared and characterized methoxy poly-(ethylene glycol)-poly(ε-caprolactone) (mPEG-PCL) nanoscale polymeric micelles carrying a combination of Sunitinib and SN-38, to examine whether these SN-38/Sunitinib co-loading polymeric micelles show significant clinical advantage and increased antitumor effects in a mouse model of colorectal cancer.

## 2. Materials and Methods

### 2.1. Synthesis of mPEG-PCL

The polymers mPEG (MW = 5000 Da) and PCL (MW = 8000 Da) were received as described elsewhere [25]. 7-ethyl-10-hydroxy-camptothecin (SN-38, ScinoPharm Ltd., Tainan city, Taiwan), and Sunitinib (Free base, LC Laboratories, Woburn, MA, USA) were also obtained.

The mPEG-PCL copolymers were synthesized; the detailed procedure has been previously described [22,26,27,28]. ^1^H nuclear magnetic resonance (NMR) (Bruker Corp., Billerica, MA, USA) was performed to quantitate the molecular weight of copolymers.

### 2.2. Preparation of SN-38 and Sunitinib Loaded mPEG-PCL Micelles

Using the lyophilization-rehydration method [25], either the SN-38 or Sunitinib was loaded into nanoscale micelles at various drug/polymer ratios (D/P ratio) of 0.25/10, 0.5/10 and 1/10, respectively. In addition, SN-38 and Sunitinib were co-loaded into micelles at D/P ratios (SN-38/Sunitinib/polymer) of 0.25/1/10, 0.25/2/10, 0.5/1/10, 0.5/2/10 and 1/1/10. SN-38, Sunitinib, SN-38/Sunitinib and the amphiphilic copolymers mPEG-PCL were dissolved in dimethyl sulfoxide (DMSO). Then sonication was performed along with preparation methods as described in previous studies [22,25].

### 2.3. Characterization

Particle size, and polydispersity index (PDI) of the SN-38 micelles, Sunitinib micelles, and SN-38/Sunitinib micelles at each D/P ratio were evaluated using a dynamic light scattering (DLS) system (Zetasizer, Nano-ZS90, Malvern Instruments Ltd., Malvern, UK) and transmission electron microscopy (TEM). The SN-38 micelles, Sunitinib micelles and SN-38/Sunitinib co-loading micelles were dissolved with DMSO. All supernatants of micelles were measured to assess the loading capacity and encapsulation efficiency of SN-38 or/and Sunitinib. The SN-38 or/and Sunitinib content was determined by HPLC (C18 column) with FL detection (ex 365 nm, em 550 nm) and UV detection (424 nm), respectively. The loading capacity and encapsulation efficiency were calculated by following equations:(1)$$\mathrm{Loading}\;\mathrm{capacity}\;\left(\%\right)=\frac{\mathrm{amount}\;\mathrm{of}\;\mathrm{drug}}{\mathrm{amount}\;\mathrm{of}\;\mathrm{polymer}+\;\mathrm{amount}\;\mathrm{of}\;\mathrm{drug}}\times 100\%$$
(2)$$\mathrm{Encapsulation}\;\mathrm{efficiency}\;\left(\%\right)=\frac{\mathrm{amount}\;\mathrm{of}\;\mathrm{drug}\;\mathrm{loading}}{\mathrm{theoretical}\;\mathrm{drug}\;\mathrm{loading}}\times 100\%$$

The drug release from 2 mL of SN-38/Sunitinib micelles was analyzed with a dialysis bag (molecular weight = 3.5 kDa) in 50 mL of PBS (Phosphate buffered saline) with 0.1% Tween 80 solution by diffusion at 37 °C, 300 rpm. At selected time points, 10 μL of solution was collected from the dialysis bag which was assessed using an ELISA (Enzyme-linked immunosorbent assay) reader equipped with a UV (Ultraviolet) detector (Absorbance of 390 nm for SN-38 and 445 nm for Sunitinib).

### 2.4. Cell Culture

Three human colorectal cancer cell lines were purchased from the American Type Culture Collection (ATCC) (Manassas, VA, USA) for in vitro and/or in vivo studies, and included HT-29, SW-620, and HCT-116. The HT-29 and SW620 cells were maintained in Dulbecco’s modified Eagle’s medium (DMEM) (SIGMA-Aldrich, MO, USA). The HCT-116 cell line was maintained in McCoy’s 5A modified medium (SIGMA-Aldrich, MO, USA). The two mediums were both combined with 10% fetal bovine serum (FBS; SAFC, MO, USA) and 1% P/S (Pen Strep; GIBCO, USA), respectively, and incubated at 37 °C with 5% CO_2_.

### 2.5. Cell Viability

The cellular viability of the pure drugs and micelles was measured by an MTT assay. In the drug combination studies, the three colorectal cancer cell lines (HT-29, SW-620, and HCT-116 cells) were seeded into 96-well plates, the cell densities were 1 × 10^4^ cells (HT-29 and SW-620) or 8 × 10^3^ cells (HCT-116) per well, respectively. At 24 h after seeding, various concentrations of SN-38 and Sunitinib were added to the medium. In the drugs and micelles cell viability studies, the HCT-116 colorectal cancer cell line was seeded into 96-well plates, and the cell density was 8 × 10^3^ cells (HCT-116) per well. At 24 h after seeding, various concentrations of pure micelles, SN-38, Sunitinib, SN-38 micelles, Sunitinib micelles and SN-38/Sunitinib micelles were added to the plates. The MTT assay was performed after 72 h of incubation, and the 96-well plates of the experiment were measured using an ELISA reader (570 nm) [22]. Then the cell viability of different treatments was calculated as the mean percentage relative to the control.

### 2.6. Cellular Uptake

For the cellular uptake study, the HCT-116 cells were seeded into a two-well chamber slide with 2 × 10^5^ cells per well and incubated for 24 h at 37 °C. The cultured cells were treated with Sunitinib micelles and SN-38/Sunitinib micelles. After treatment, all the cells were imaged under a confocal microscope (Zeiss, Jena, Germany) after Hoechst staining [25].

### 2.7. Animal Model

An HCT-116 xenograft subcutaneous colorectal cancer was generated in BALB/c nude (BALB/cAnN.Cg-Foxn1nu/CrlNarl) mice. Female BALB/c nude mice of 5–6 weeks of age were purchased from the National Laboratory Animal Center (Taipei, Taiwan) and bred in the animal facility of INER (Taoyuan, Taiwan). HCT-116 cells (2 × 10^6^) were inoculated subcutaneously into the mice. All the animal procedures were performed following approved protocols that were developed in accordance with recommendations for the proper use and care of laboratory animals (Approval Number: 109006). All mice were kept at 21–23 °C in a light-dark cycle of 12 h.

### 2.8. Ex Vivo Biodistribution

The ex vivo biodistribution of Sunitinib, Sunitinib micelles and SN-38/Sunitinib co-loading micelles were studied using green fluorescence of the Sunitinib itself in mice bearing HCT-116 tumors. The subcutaneous HCT-116 tumor bearing mice were sacrificed at 48 h after treatment with injected Sunitinib solution (equivalent to 10 mg/kg of Sunitinib with 1% DMSO), Sunitinib micelles (equivalent to 10 mg/kg of Sunitinib) or SN-38/Sunitinib co-loading micelles (equivalent to 2.5 mg/kg of SN-38 and 10 mg/kg of Sunitinib), after which the main organs (heart, liver, spleen, lung and kidney) and the tumor tissue(s) were imaged using the Xtreme living image system (Bruker; Manchester, UK).

### 2.9. Antitumor Efficacy

Treatment was initiated in mice bearing the HCT-116 tumors when the subcutaneous tumor volume was greater than 50 mm^3^. The subcutaneous tumor mice were randomly allocated to the following groups (*n* = 3–4): control (normal saline), Sunitinib micelles (equivalent to 10 mg/kg of Sunitinib), SN-38 micelles (equivalent to 2.5 mg/kg of SN-38), or SN-38/Sunitinib micelles (equivalent to 2.5 mg/kg of SN-38 and 10 mg/kg of Sunitinib). All mice in each group received six doses of the pharmaceutical agents or micelles by intravenous injection every 3 to 4 days. The tumor sizes and body weights of HCT-116 tumor bearing mice were measured every 3 to 4 days during the experimental period which ended 28 days after initial treatment. In addition, tumor volume was calculated as 0.5 × (length of the tumor × (width of the tumor)^2^). The tumor growth inhibition percentage (TGI%) was also measured at the end of observation period.

### 2.10. Histochemical Staining

To evaluate tissue cytotoxicity, treatment effects, and tumor-associated fibrosis, three mice per group with mice bearing subcutaneous HCT-116 tumors were sacrificed at day 28 after injection with normal saline, SN-38 micelles (equivalent to 2.5 mg/kg of SN-38), Sunitinib micelles (equivalent to 10 mg/kg of Sunitinib), or SN-38/Sunitinib micelles (equivalent to 2.5 mg/kg of SN-38 and 10 mg/kg of Sunitinib), and the main organs (heart, liver, spleen, lung and kidney) and tumor tissues of experiment groups were obtained. In the histochemical staining study, the tumor tissues slices were stained with hematoxylin and eosin (H&E) stain.

### 2.11. Effect of Sunitinib Inhibition on Multi-Receptors

As one of the functions of Sunitinib is to target and inhibit VEGF receptor processing, mice with mice bearing subcutaneous HCT-116 tumors were sacrificed at 48 h after treatment. Mice with injected normal saline, SN-38 micelles (equivalent to 2.5 mg/kg of SN-38), Sunitinib micelles (equivalent to 10 mg/kg of Sunitinib), or SN-38/Sunitinib co-loading micelles (equivalent to 2.5 mg/kg of SN-38 and 10 mg/kg of Sunitinib), were injected with dextran (1 mg/100 uL, intravenous injection) 10 min before sacrifice and the tumor tissues were harvested for fluorescence imaging (BX-51; Olympus, Tokyo, Japan).

For the Sunitinib inhibition of PDGF receptors study, the tumor tissues slices obtained as described in the above histochemical staining study were stained with Sirius red stain (Picro Sirius red stain Kit, Connective Tissue Stain, abcam, Cambridge, UK), and five Sirius red images obtained (200×) from each of the four treatment groups were quantified using Image J (NIH, Bethesda, MD, USA).

### 2.12. Statistical Analysis

All data of this research were presented as mean ± standard deviation, and the *t*-test was used to test for significant differences between groups. Values of *p <* 0.05 or *p <* 0.01 were considered statistically significant.

## 3. Results

### 3.1. Synthesis and Characterization

The copolymers of mPEG-PCL were prepared, and characterized by ^1^H NMR (Appendix A). The results of DLS demonstrated that the SN-38 micelles in the D/P ratios of 0.25/10, 0.5/10 and 1/10 were 271.5 nm, 268.1 nm and 163.6 nm, respectively. The PDI of SN-38 micelles at a D/P ratio of 1/10 was 0.150 (Appendix A), as previously reported [22]. Sunitinib micelles at D/P ratios of 0.25/10, 0.5/10 and 1/10 were 97.5 nm, 99.2 nm and 97.7 nm, respectively, while the PDI of Sunitinib micelles at a D/P ratio of 1/10 was 0.109. The loading capacity and encapsulation efficiency of Sunitinib were 4.76% and 50.02%, respectively. The PDI of SN-38/Sunitinib micelles at a D/P ratio of 1/1/10 was 0.134. The loading capacity and encapsulation efficiency of SN-38 were 6.65% and 71.18%, respectively. Furthermore, SN-38/Sunitinib co-loading micelles in each D/P ratio of 0.25/1/10, 0.25/2/10, 0.5/1/10, 0.5/2/10 and 1/1/10 were 89.2 nm, 92.3 nm, 88.8 nm, 92.4 nm and 112.8 nm, respectively. The PDI of SN-38/Sunitinib micelles at a D/P ratio of 0.25/2/10 was 0.116. The loading capacity and encapsulation efficiency of SN-38 were 2.39% and 98.70%, respectively, and those of Sunitinib were 14.73% and 86.41%, respectively. The SN-38 micelles, Sunitinib micelles and SN-38/Sunitinib micelles at D/P ratios of 1/10 and 0.25/2/10 were chosen for the following experiments.

As shown in Figure 1, TEM was used to characterize the mPEG-PCL micelles, SN-38 micelles, Sunitinib micelles, and SN-38/Sunitinib micelles, which revealed a spherical morphology with similar particle sizes as measured by DLS (Appendix A).

The FTIR spectra of SN-38/Sunitinib micelles, Sunitinib, mPEG-PCL and SN-38 were scanned within 550–4000 cm^−1^ as shown in Appendix A. (Acquired by PerkinElmer Spectrum 100, Waltham, MA, USA). The characteristic peaks of SN-38 at 1170 cm^−1^ (C–O stretch), 1733 cm^−1^ (C=O stretch) and 3583 cm^−1^ (O–H stretch) and Sunitinib at 2969 cm^−1^ were present in the spectra of the SN-38/Sunitinib micelles. These results indicate that SN-38, Sunitinib or SN-38/Sunitinib was indeed incorporated into the micelles.

Next, the release of SN-38 or Sunitinib from SN-38/Sunitinib co-loading micelles was evaluated. Appendix A shows the rate of SN-38 or Sunitinib release from the co-loaded micelles. Within 72 h, about 35% of the SN-38 and 85% of the Sunitinib had been released from the SN-38/Sunitinib micelles.

### 3.2. Cell Viability

In vitro drugs combination cytotoxicity assays indicated that SW-620, HT-29, and HCT-116 cells had lower cell viability after treatment with combined SN-38 and Sunitinib (Figure 2). After treatment, HCT-116 cell lines had higher cell death compared to SW-620 and HT-29 cells (*p* < 0.05). In vitro drugs and micelles cytotoxicity assays indicated that HCT-116 cells had low cell viability after treatment with SN-38/Sunitinib micelles (Figure 3). In contrast, treatment of the HCT-116 cells with Sunitinib demonstrated higher cell viability. Pure micelles (mPEG-PCL micelles) did not show any cytotoxicity (average cell viability > 100%), whereas the SN-38/Sunitinib co-loading micelles showed a highly therapeutic effect at 72 h after treatment (*p* < 0.01), compared with pure micelles and Sunitinib micelles treatment groups.

### 3.3. Cellular Uptake

The cellular uptake of the drugs in HCT-116 cells treated with Sunitinib micelles, and SN-38/Sunitinib micelles was assessed after 24 h at 37 °C. As shown in Figure 3, HCT-116 cells incubated with Sunitinib micelles (Figure 4A) and SN-38/Sunitinib micelles (Figure 4B) both showed stronger green fluorescence of Sunitinib at 24 h after treatment. However, cellular uptake of Sunitinib indicated that it was not present in the cell’s nucleus.

### 3.4. Ex Vivo Biodistribution

The heart, liver, spleen, lung, kidneys, and tumor tissues were isolated to evaluate the ex vivo bioluminescent presence of Sunitinib, Sunitinib micelles, and SN-38/Sunitinib micelles using green fluorophore of Sunitinib at 48 h after injection into mice with HCT-116 tumors (Figure 5). The accumulation images show that the tumor of the mice had higher concentration of Sunitinib, Sunitinib micelles and SN-38/Sunitinib micelles, compared with other tissue organs. In addition, the Sunitinib micelles and SN-38/Sunitinib micelles had a higher accumulation of Sunitinib compared with Sunitinib only.

### 3.5. Antitumor Efficacy

For antitumor efficacy studies, we used mice with subcutaneous HCT-116 colorectal tumors, which were established at 7–10 days after injection (Figure 6). The average tumor volume was 151 ± 75 mm^3^ on day 0, and average mice body weight was 23 ± 3 g. We expected that the SN-38/Sunitinib co-loading micelles (2.5 mg/kg of SN-38 and 10 mg/kg of Sunitinib) would reduce tumor growth, and the reduced dosage of SN-38 provides a similar antitumor effect on colorectal cancer compared with our previous studies [22]. Tumor volume in SN-38/Sunitinib micelles treated mice was 118 ± 84 mm^3^ at 28 days after the initial treatment, and was significantly different from tumor volumes in animals treated with SN-38 micelles (2495 ± 612 mm^3^, *p* < 0.01). The tumor sizes in normal saline treated animals were all above 3000 mm^3^ (*p* < 0.01; Figure 6A). These results imply that the largest antitumor effect was achieved when low-dose SN-38 (2.5 mg/kg) was combined with high-dose Sunitinib (10 mg/kg) in co-loading micelles. Furthermore, these findings also demonstrate that Sunitinib might be a promising strategy for adjuvant chemotherapy in colorectal cancer, as reported by previous studies [29,30]. In addition, the body weight of mice treated with SN-38/Sunitinib micelles had not significantly reduced at day 28, compared with normal saline, SN-38 micelles or Sunitinib micelles treated mice (*p* > 0.05) (Figure 6B).

At 28 days after initial treatment, TGI% values in mice with HCT-116 tumors treated with SN-38/Sunitinib co-loading micelles and SN-38 micelles were 97% and 48%, respectively.

### 3.6. Histochemical Staining

The histochemical analysis of normal organs showed no significant toxicity in relation to the treatments (Appendix A). As shown in Figure 7, the HCT-116 tumors treated with SN-38/Sunitinib micelles had the largest areas of necrosis and apoptosis, compared to the control, SN-38 micelles, and Sunitinib micelles groups.

### 3.7. The Effect of Sunitinib on Tumor-Associated Vasculature and Fibrosis

We assessed the vascular distribution and Sirius red stain of HCT-116 tumors after treatment. As seen in Figure 8, the tumor vasculature in mice treated with normal saline or SN-38 micelles was structurally abnormal. In contrast, the normalization of tumor vasculature was observed in mice treated with Sunitinib micelles or SN-38/Sunitinib micelles.

The results of tumor-associated fibrosis by Sirius red stain (Figure 9) in HCT-116 tumors showed that SN-38/Sunitinib co-loading micelles significantly reduced tumor-associated fibrosis after treatment compared with the control, SN-38 micelles and Sunitinib micelles (*p* < 0.01).

## 4. Discussion

The characterization studies demonstrated that the average particle size of SN-38/Sunitinib micelles was less than 100 nm (Appendix A and Figure 1), which could enhance accumulation in the tumor by the Enhanced Permeability Retention (EPR) effect [22,31,32].

In the release study (as shown in the Appendix A), the release rates of SN-38 or Sunitinib from the co-loaded micelles were about 35% of the SN-38 and 85% of the Sunitinib within 72 h. The release profile proved that SN-38/Sunitinib micelles could release drugs in the cancer tissues for therapy [33,34,35].

In the drug combination studies, we demonstrated that the combined treatment of SN-38 and Sunitinib could inhibit cell viability. Three colorectal cancer cell lines (HT-29, SW-620, and HCT-116 cells) were studied. The results show HCT-116 cell lines had higher cell death compared with other cell lines (*p* < 0.05) (Figure 2). Then, the results of the drug combination studies suggest that the HCT-116 cell line had less drug resistance to SN-38 and Sunitinib. In vitro cytotoxicity (Figure 3) showed HCT-116 cells had low cell viability after treatment with SN-38 as reported previously [22], which is consistent with claims that SN-38 is a highly promising chemotherapeutic agent [36,37,38]. Sunitinib demonstrated higher cell viability that was consistent with previous reports that had indicated that Sunitinib can change the tumor microenvironment, and weaken ability to inhibit cell growth [13,39,40]. In vitro cell viability studies on HCT-116 cells showed that the nanoscale micelles could deliver the drugs into cancer cells and increase cytotoxicity. The cell viability results showed that SN-38 combined with Sunitinib had divergent cell cytotoxicity profiles at different concentrations. SN-38/Sunitinib co-loading micelles showed a highly therapeutic effect.

Results of cellular uptake (Figure 4) were in accordance with other published reports that demonstrated drug presence in the endosome and/or lysosome of the cell [41,42,43,44]. There were fewer cells in the combination SN-38/Sunitinib (Figure 4A) and SN-38/Sunitinib micelles (Figure 4B) since SN-38 has high cell cytotoxicity.

The accumulation images (Figure 5) demonstrated that the lungs of the mice had higher concentration of Sunitinib, Sunitinib micelles and SN-38/Sunitinib micelles, as found in several previous reports [45]. Additionally, less accumulation was found in all kidneys, indicating that a little Sunitinib is metabolized by the kidney. The tumors had a higher accumulation after the injection of Sunitinib micelles and SN-38/Sunitinib micelles (Figure 5C) compared with Sunitinib only (Figure 5A). The liver did not have any relative fluorescence intensity after injection of Sunitinib, Sunitinib micelles or SN-38/Sunitinib micelles, implying that the micelles might be filtered by the tissue. Thus, our results showed high Sunitinib accumulation in tumors from the injected SN-38/Sunitinib micelles by the EPR effect, which would, theoretically, enhance the therapeutic antitumor effect.

In antitumor efficacy studies, Sunitinib micelles demonstrated poor tumor inhibition (Figure 6A). Taken together, all the above results demonstrated that treatment with dual-loaded SN-38/Sunitinib micelles could significantly improve the treatment effect (*p* < 0.01; TGI% was 97% at 28 days post injection compared with other treatment groups. The antitumor effect studies results confirmed that SN-38/Sunitinib micelles could be used in the treatment of colorectal cancer bearing mice. The body weight of mice treated with SN-38/Sunitinib micelles had not significantly reduced at day 28 after injection (Figure 6B). Mice of the SN-38 micelles group had lower body weight during the observation period, but this was not significantly different from body weights of other treatment groups (*p* > 0.05).

Mice bearing HCT-116 tumors treated with SN-38/Sunitinib micelles (Figure 7D) demonstrated the largest areas of necrosis and apoptosis by H&E staining compared with the control and other treatment groups (Figure 7). The results of tissue toxicity and antitumor effects suggest that the SN-38/Sunitinib micelles had the largest response and were clearly safer (Appendix A). Histochemical analysis of major tissues and tumors by H&E staining was used to evaluate the tissue toxicity and antitumor effects of SN-38/Sunitinib co-loading micelles. The ex vivo tissue distribution and anti-tumor studies demonstrated that nanoscale micelles significantly improved the tumor accumulation and therapeutic efficacy, and reduced side effects on normal tissues. All the results conformed with the release study; the SN-38 and Sunitinib of SN-38/Sunitinib micelles could be released from mPEG/PCL and exert their effects.

Sunitinib is a multi-target RTKI that can target the vascular endothelial growth factor receptors (VEGF-1, VEGF-2, VEGF-3 receptors) and the platelet-derived growth factor receptors (PDGF-α and PDGF-β receptors). This inhibitory effect is related to tumor-vasculature normalization (VEGF receptors) and tumor-associated fibrosis (PDGF receptors) [46,47], respectively. The anti-angiogenic effect of Sunitinib micelles (Figure 8C) and SN-38/Sunitinib micelles (Figure 8D) initially alleviated these abnormalities by inhibiting angiogenesis [48]. The results (Figure 8) also demonstrated that Sunitinib provided pharmacodynamics evidence of vascular normalization. PDGF expression and action have been reported to be involved in tumor-associated fibrosis [46], and the Sirius red stain is usually used to evaluate fibrosis in tissue (Figure 9) [47]. Moreover, the effect of Sunitinib inhibition on multi-receptor studies of SN-38/Sunitinib micelles (Figure 9E) also demonstrated that the Sunitinib blocks the activation of VEGF and PDGF receptors. Therefore, we consider that dual-loaded SN-38/Sunitinib micelles probably inhibit tumor cell proliferation through the influence of SN-38 and also inhibit angiogenesis (VEGF receptor) and reduce tumor-associated fibrosis (PDGF receptor) through the influence of Sunitinib.

## 5. Conclusions

In conclusion, we prepared and characterized dual-loaded SN-38/Sunitinib nanoscale micelles. In in vitro and in vivo studies, the nanoscale SN-38/Sunitinib micelles could efficiently deliver the drugs into cancer cells, thereby increasing their cytotoxicity, and significantly improved their tumor accumulation and antitumor efficacy. Thus, we consider that dual-loaded SN-38/Sunitinib nanoscale polymeric micelles to be a treatment modality that inhibits VEGF and PDGF receptors and enhances the chemotherapeutic effect in a mouse model of colorectal cancer.

## Data Availability

The data presented in this study are contained within the article.

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
