# Peer review of "Dual-Functional Polymeric Micelles Co-Loaded with Antineoplastic Drugs and Tyrosine Kinase Inhibitor for Combination Therapy in Colorectal Cancer"

_pharmaceutics, 2022, doi:10.3390/pharmaceutics14040768_

Round 1

Reviewer 1 Report

The experimental study itself is interesting in how it investigates the combinational effects of two drugs to form a synergy that has a better therapeutic response. However, unfortunately, there are many issues with this manuscript that prevent its acceptance in its current form. 

This is not novel work, as there have been other studies in the past and clinical trials investigating this combination (Cancer Res (2012) 72 (8_Supplement): 4375. https://doi.org/10.1158/1538-7445.AM2012-4375)

The authors base all of their work and hypothesis on a single colorectal cancer cell line. Additional cell lines are needed to truly make conclusions about colorectal cancer - perhaps the cell line chosen is more sensitive than other colorectal lines. More evidence is needed. 

I recommend that the authors use professional software to generate "Scheme 1." In its current state, some of the text are blurry and aspects of the figure are not to scale (why are micelles as large as the mouse in the image)? 

Figure 2 is not convincing enough regarding the SN38-Sunitinib micelles combination. It is difficult to distinguish whether the decrease in viability is only due to SN-38 alone or not. It is also difficult to determine whether these increased concentrations are toxic to the cells in general, or whether they actually have a therapeutic effect. 

Lastly, the writing needs to be significantly improved throughout the manuscript. For example, the first sentence in the intro "CRC is the third most common cancers in the worldwide," is not written correct - it should be CRC is the third most common cancer in the world. I also noticed a couple of spelling mistakes. 

Author Response

Appendix is our point-by-point responses to the reviewer’s comments and suggestions.

Reviewer 2 Report

The manuscript is well presented and written. I only have two comments:

  • Figure 2. Does the concentration refer to the micelles or the drugs? Did you calculate the IC50? Did you measure the combination index of these drugs? 
  • Why was a subcutaneous tumor tested? Because for a subcutaneous tumor you can inject intratumoral, there is no need to inject IV.

Author Response

(The authors gave the same response as above.)

Reviewer 3 Report

The article entitled Dual‐functional polymeric micelles co‐loaded with antineo

 plastic drugs and tyrosine kinase inhibitor for combination  therapy in colorectal cancer is a document of interesting subject matter.

However, it needs some major changes before being accepted. Make the following corrections:

  1. Introduction is short and weak. Please improve introduction by introducing current challenges by standard medical to treat colorectal cancer and then introducing nanotechnology to combat with current challenges. Please cite to the paper in following:
  • DOI: 10.1007/s11051-020-05129-6
  • DOI: 10.1016/j.ijbiomac.2021.12.052
  • DOI: 10.1007/s12247-022-09621-5
  • DOI: 10.1016/j.jddst.2022.103138
  • DOI: 10.3390/app12010477
  • DOI: 10.3390/pharmaceutics14030472
  1. Please pay attention on more interpretation of the experimental results related to TEM characterization.

Line 185: authors regarding morphology nanomicelles stated “As shown in Figure 1, TEM was used to characterize the mPEG-PCL micelles, SN-micelles, Sunitinib micelles, and SN-38/Sunitinib micelles, which revealed a spherical morphology with similar particle sizes measured by DLS”, but all of them are not spherical. Please more clarify in this case.

  1. Also, there are aggregates for some of nanomiclles, but no discussion on this issue.

No effect on invitro and invivo studies?

  1. In this work, nanomicelles containing drug synthesized. In other words, our samples are liquid samples and not solid samples. With this in mind, authors how to get nanomicelle size by an ordinary TEM tool due high vacuum. Please more clarify in this case.

For characterization the exact size of liquid samples, it is needed to having Cryo-TEM.

  1. Please try to more discussion on relationship between the nanomicelle size AND invitro / invivo studies of micelles.
  2. Any try to do the invitro / invivo studies of micelles without drug?
  3. The authors should cite and discuss some related studies about these nanomicelles especially in EE% and kinetics evaluation of the release study of drug from the nanomicelles.

  1. The conclusion is a bit too concise. Please make a general conclusion of the study.
  2. The discussion in the release study is weak. Please compare the results of your paper with another similar study.
  3. I recommend the inclusion of a scheme on preparation of SN-38 and Sunitinib loaded mPEG-PCL micelles.

Author Response

(The authors gave the same response as above.)

Round 2

Reviewer 3 Report

It is acceptable now.